# Medium Cut-Off Dialysis Membrane and Dietary Fiber Effects on Inflammation and Protein-Bound Uremic Toxins: A Systematic Review and Protocol for an Interventional Study

**DOI:** 10.3390/toxins13040244

**Published:** 2021-03-29

**Authors:** Tjaša Herič, Tjaša Vivoda, Špela Bogataj, Jernej Pajek

**Affiliations:** 1Department of Nephrology, University Medical Centre Ljubljana, 1000 Ljubljana, Slovenia; tjasa.heric@kclj.si (T.H.); tjasa.vivoda@kclj.si (T.V.); spela.bogataj@kclj.si (Š.B.); 2Department of Internal Medicine, Faculty of Medicine, University of Ljubljana, 1000 Ljubljana, Slovenia

**Keywords:** protein-bound uremic toxins, *p*-cresyl sulfate; indoxyl sulfate, inflammation; IL-6, medium cut-off membrane, dietary fiber supplementation

## Abstract

The aim of this systematic review is to investigate the effects of the use of a medium cut-off membrane (MCO) and dietary fiber on the concentration of protein-bound uremic toxins (PBUTs) and inflammatory markers in hemodialysis (HD) patients. Of 11,397 papers originally found, eight met the criteria of randomized controlled trial design. No study examined the effects of MCO membranes on PBUTs. Three studies examined the reduction in inflammatory markers with MCO membranes compared to high-flux HD membranes and showed no significant differences. Five studies of dietary fiber supplementation showed an inconclusive positive effect on PBUT levels and a significant positive effect on the reduction in inflammatory markers (interleukin-6 reduction: standardized difference in means −1.18; 95% confidence interval −1.45 to −0.9 for dietary fiber supplementation vs. control; *p* < 0.001). To date, no study has combined the use of an MCO membrane and fiber supplementation to reduce PBUT levels and inflammation with online hemodiafiltration as a comparator. A rationale and protocol for an interventional trial using a combination of MCO membrane dialysis and fiber supplementation to lower inflammatory markers and PBUT concentrations are presented.

## 1. Introduction

Uremic syndrome is caused by the progressive retention of numerous uremic toxins (UTs) in patients with end-stage kidney disease (ESKD). UTs are classified into three main groups: free water-soluble low molecular weight solutes, medium and large molecular weight solutes and protein-bound solutes [1]. Most UTs have negative effects on cardiovascular, inflammatory and fibrogenic systems and may importantly contribute to high morbidity and mortality in chronic kidney disease (CKD) [2]. The clearance of some middle–large uremic toxins is insufficient by modern hemodialysis methods, while protein-bound uremic toxins (PBUTs) are even more difficult to eliminate due to their strong binding to serum proteins, with only a small fraction of the unbound solutes being removed by dialysis [3]. It is quite possible that a significant portion of the adverse effects of the residual uremic syndrome observed in today’s dialysis population is the result of the inadequate clearance of these larger and poorly eliminated PBUTs. Important determinants of the concentration of medium–large uremic toxins and PBUTs are not the dose of dialysis (measured by urea Kt/V) but mainly the residual renal function and the composition of the diet [4]. There are two possible strategies for improving the removal of these toxins: first, increasing clearance by improving dialysis technologies and membranes, and second, reducing the formation and/or absorption of various toxins and toxin precursors.

Online hemodiafiltration, currently considered the best dialysis method for chronic renal replacement therapy, uses high convective volumes for the efficient removal of middle molecules and provides better results compared to high-flux dialysis membranes [5,6,7,8]. High-flux dialysis membranes used in high-flux hemodialysis and online hemodiafiltration provide the better clearance of middle molecules (e.g., β2 microglobulin). However, their cut-off values up to about 20 kDa [5,9] do not provide sufficient clearance to significantly reduce the concentrations of larger uremic toxins. They may improve the survival of patients when treated with large filtration volumes [5,6]. Larger middle molecules, sized > 25 kDa, need to be removed by higher convection rates or by using highly permeable membranes. Recently, a new concept called expanded hemodialysis (HD) (HDx) was developed [9]. HDx is defined as a treatment in which diffusion and convection are conveniently combined inside a hollow fiber dialyzer equipped with a medium cut-off membrane (MCO) [10]. This latest generation of dialysis membranes has been developed to achieve filtration close to that of the natural kidney (i.e., to mimic the filtration ability of the glomerular membrane). Compared to high-flux membranes, MCO membranes have slightly larger pores with tight pore distribution, higher hydraulic water permeability and thus good permeability for larger solutes (15,000–45,000 Da), and at the same time, low permeability for albumin [9,10,11], making them suitable for patients undergoing chronic maintenance hemodialysis.

When considering possible strategies to reduce PBUT generation and absorption, it is important to note that many PBUTs are produced by the metabolism of food constituents in the gut microbiota, mainly amino acid metabolites, such as indoles and phenols [3]. The best studied representatives of this group are *p*-cresyl sulfate (PCS) and indoxyl sulfate (IS). They are produced by the bacterial fermentation of food protein in the colon. PCS is the metabolite of amino acids tyrosine and phenylalanine, and IS of tryptophan. Both are 90% bound to serum albumin, which explains their poor removal by dialysis. They are mainly excreted in the urine; therefore, their serum concentration is significantly increased in ESKD. With increasing concentration, binding saturation occurs, which leads to an increased free fraction of PCS and IS, which can be further increased in ESKD patients by a low albumin concentration [12]. The production of these toxins is increased in uremia due to shifts in the microbiota in favor of proteolytic to saccharolytic microorganisms. As a result of reduced removal and increased production, the circulating levels of PCS and IS in HD patients can be up to 30 times higher than in healthy individuals [13,14]. PCS and IS are associated with cardiovascular damage, cardiovascular events, increased mortality in CKD, kidney damage and CKD progression [15,16,17]. In general, the higher the protein intake or protein to fiber ratio in the diet, the higher the concentration of PBUTs [4,14]. On the other hand, the higher the dietary fiber intake, the lower the concentration of PBUTs [18,19,20]. Increased dietary fiber intake was associated with reduced inflammation and oxidative stress in HD patients [21,22]. The proposed mechanisms through which increased fiber intake modifies PBUT and inflammation are decreased gut transit time, increased bacterial formation of short-chain fatty acids with improved gut mucosal barrier function, reduced bacterial product translocation and reduced PBUT formation by reduced proteolytic bacterial fermentation [23].

In vitro studies [24,25,26,27] and studies in CKD patients [28] have suggested an association of some PBUTs with inflammatory and oxidative stress markers. Rossi et al. [28] demonstrated a significant correlation of free and total IS with interleukin-6 (IL-6), interferon-gamma (IFN-g) and tumor necrosis factor-alpha (TNF-a), as well as a correlation of free and total PCS with IL-6. In vitro studies showed the ability of PCS and IS to stimulate the gene expression of IL-6 [24] and IS to stimulate the expression of TNF-a [25]. PCS and IS induced reactive oxygen species, which activate the nuclear factor-kappaB (NF-kb) signaling pathway, leading to oxidative stress as well as the production of proinflammatory cytokines [26,27]. In addition, the inflammatory effects of PCS and IS are related to free radical production and fibrosis [29]. PCS and IS are both associated with renal tubulointerstitial fibrosis and cardiovascular disease progression [29,30]. IS has been associated with smooth muscle cell proliferation and vascular calcification, and thus with the progression of atherosclerosis with increased aortic stiffness and the progression of peripheral arterial disease. Through the mechanism of cardiac hypertrophy and fibrosis, increased IS levels have been associated with congestive heart failure and arrhythmia in CKD patients [31,32,33]. IS has been associated with thrombosis in vascular accesses in HD patients [31] and has also been associated with disorders in the hemostatic system [34] and renal bone disease in CKD patients [32,35]. Elevated PCS has been associated with insulin resistance and the development of metabolic syndrome and its complications in CKD patients [30]. All in all, these data show a plethora of evidence for the influence of PBUT on various factors leading to increased morbidity in dialysis patients.

Here, we present a systematic review of MCO membranes and dietary fiber usage in the field of chronic renal replacement therapy on PBUT concentrations and inflammation levels in HD patients. Next, we present a rationale and protocol for an interventional study combining both approaches to reduce these endpoints.

## 2. Results

### 2.1. Performance of Medium Cut-Off Dialyzers in the Elimination of Middle Molecules and Protein-Bound Uremic Toxins Compared to Hemodiafiltration or High-Flux Hemodialysis

Our search strategy generated 4882 potentially relevant articles. Furthermore, 4879 papers were excluded based on the title and abstract, as they did not meet the inclusion criteria due to the type of article, study design or outcome of interest. We reviewed three articles for eligibility, all of which (Table 1) were included in the final review (one randomized controlled trial and two randomized controlled cross-over studies). 

The summarized effects of four studies (Figure 1) showed no statistically significant difference between the interventional (MCO membrane) and control treatments (standardized difference of mean values: −0.03; 95% confidence interval: −0.23 to 0.17; *p* = 0.762).

The analysis of the studies (Table 1, Figure 1) showed predominantly equivocal results, which may show numerically lower values of IL-6 when using the MCO membrane, but the change was not statistically significant compared to high-flux hemodialysis. The reduction in other inflammatory markers (e.g., TNF-α, CRP, IFN-γ and SAA) did not differ significantly with an MCO or a high-flux dialyzer. Only one study reported superior intragroup removal of TNF-α using an MCO membrane, but the difference to controls did not reach significance. Interesting results were reported in a study by Zickler et al. [37], which showed a statistically significant lower TNF-α mRNA and IL-6 mRNA expression in peripheral blood mononuclear cells after MCO membrane dialysis compared to high-flux dialysis. The latter results suggest that the MCO dialyzer might influence the gene expression of immunoregulatory leukocyte subsets, since most of the major inflammatory cytokines were downregulated by lymphocytes and monocytes. Studies conducted to date have several limitations, including a small sample size and a short study duration. A comparison of MCO hemodialysis with hemodiafiltration is needed, since hemodialfiltration would enable the greater removal of pro-inflammatory middle molecules than high-flux hemodialysis. Therefore, further studies are needed to evaluate the ability of the MCO membrane to reduce chronic low-grade inflammation in hemodialysis patients.

### 2.2. Impact of Increased Dietary Fiber Supplementation on Serum Concentration of PCS, IS and Inflammatory Markers

Our search strategy yielded 3778 potentially relevant articles. Furthermore, 3769 articles were excluded based on the title and abstract, which clearly indicated that they did not meet the inclusion criteria. Four studies did not satisfy our criteria for inclusion. One study on HD patients was excluded because it was non-randomized. The other three were excluded because the studied population was CKD patients; two of them were also retrospective cross-sectional studies. Finally, only five studies met the inclusion criteria (Table 2).

The summarized effects of four ESs (Figure 2) showed a statistically significant impact of fiber supplementation on IL-6 levels compared to the control treatments (standardized difference of mean values: −1.18; 95% confidence interval: −1.45 to −0.9; *p* < 0.001).


Table 2 shows that fiber supplementation had a heterogeneous effect on PBUT levels, with studies showing a decrease in either PC or IS with some heterogeneity in terms of the reduction in free or total fraction. On the other hand, the reduction in inflammation was uniform, with all studies showing a decrease at IL-6 and some studies showing a decrease at other markers (TNF-a, hs-CRP). It is possible that the critical variable is perhaps not only the absolute fiber intake but also a protein/fiber ratio in the diet [14], and this may explain some of the discrepancies observed. Nevertheless, these data from a limited number of studies support the assumption of a beneficial effect of dietary fiber supplementation, which should be further investigated.

### 2.3. Quality Assessment

Two authors (T.V. and T.H.) independently performed the search, selection and evaluation of methodological quality, with a discussion and consensus (J.P.) on all observed differences. PEDro scores ranged from 5 to 9 out of 10, with a median of 7.0 (Table 3). The inter-rater reliability between the two authors was almost perfect (Cohen’s kappa = 0.92). Three studies blinded the assessors, four used intention-to-treat analysis and two concealed allocation.

## 3. Discussion and an Interventional Trial Proposal

The above analysis showed that there is some evidence that fiber supplementation has a potentially positive effect on PBUT levels and inflammation. We could not find any studies investigating the effect of MCO membranes on PBUT values. The effects of MCO membranes on inflammation markers were inconsistent and inconclusive. We found some evidence from recent studies on the use of fiber supplementation to reduce inflammation levels and PBUT concentrations. Furthermore, both interventions are generally safe, and no major adverse effects were reported in the studies reviewed. However, these findings need to be interpreted in light of significant limitations as follows.

First, the number of available and eligible randomized controlled studies is very limited, and the total number of included patients is low. Therefore, this makes our findings susceptible to random effects. Second, as shown in Table 3, in some of the studies, intention-to-treat analyses were not included, outcome assessors were not blinded, there was no concealed allocation and interventions and outcome measures were poorly reported. Since we made no attempt to identify unpublished registered clinical trials, we cannot exclude the possibility of publication bias. In addition, most studies did not calculate the required sample size for accurate effect estimation. Therefore, additional studies are needed to extend the current limited evidence and to address the above limitations.

Against this background, it is a natural step to combine the use of a novel purification therapy (the MCO membrane) and the use of a cheap, easily accessible and adverse effect-free intervention to reduce PBUT generation (dietary fiber supplementation) and to reduce the two important components of residual uremic syndrome—PBUT concentration and inflammation. This proposal could be further supported by a possible synergistic effect of both interventions, as there are known links between PBUTs and inflammation induction (see introductory section above). Additionally, the uremic state per se may induce an increase in PBUT generation due to a modification of the microbiota [39]. Therefore, we speculate whether the use of a medium-cut off dialysis membrane and dietary fiber supplementation could act synergistically and induce a significant and clinically meaningful reduction in inflammation levels and PBUT concentration compared to online hemodiafiltration with a high-flux dialysis membrane.

On this basis, we designed a prospective, randomized, controlled and interventional study in two parallel groups to compare the serum concentrations of selected protein-bound uremic toxins and inflammatory markers obtained by medium cut-off membrane dialysis (Theranova, Baxter AG, USA) and online hemodiafiltration with a standard high-flux dialysis membrane. In the extension phase, patients in both study arms will undergo a dietary change with an increase in daily fiber intake and the addition of a short-chain fatty acid propionate. The two main outcomes of the study will be the serum concentration of *p*-cresol sulfate and the serum concentration of IL-6. Secondary outcomes will include the serum concentration of indoxyl-sulfate, trimethylamine-N-oxide (TMAO), IL-10, serum amyloid A (SAA), high-sensitivity C-reactive protein (hs-CRP), total leukocyte count and plasma concentration of bacterial 16s rDNA. The serum albumin concentration and lean body weight of patients will represent safety outcomes for this study.

This investigator-initiated research project has been entirely planned and will be conducted by the clinical researchers in a tertiary hospital, University Medical Centre Ljubljana, Department of Nephrology. The study will be conducted in accordance with the ethical principles of the Declaration of Helsinki and was approved by The National Medical Ethics Committee of the Republic of Slovenia (numbers 0120-430/2019/12, date 22 October 2019 and 0120-608/2019/3, date 23 January 2020). All study participants will have to provide informed written consent prior to study enrolment.

The trial will enroll 50 chronic prevalent clinically stable HD patients in the following periods:

First two weeks of the wash-in period with standard bicarbonate hemodialysis and standard high-flux dialysis membrane. Afterwards, the patients will be randomized in a 1:1 ratio to either one of the two study groups:Experimental group: 4 weeks of dialysis with medium cut-off (Theranova) membrane (first phase), and dialysis for 8 weeks with the same membrane and increased fiber and sodium propionate intake (second phase);Control group: 4 weeks of dialysis with a high-flux membrane using online hemodiafiltration (first phase) and 8 weeks of high-flux membrane hemodiafiltration and increased fiber and sodium propionate intake (second phase).

Finally, all patients will undergo a four-week wash-out period with standard bicarbonate hemodialysis using a standard high-flux dialysis membrane identical to the wash-in period.

Measurement of the study endpoints will be carried out at the end of the 2-week wash-in period (baseline), after each interventional period (4-week; 8-week), and at the end of the 4-week wash-out period. The flow diagram of the study is presented in Figure 3.

When determining the duration of wash-in and wash-out periods, we considered the half-life of uremic toxins and inflammation markers, e.g., the half-life for IL-6 is in the range of 2–15 h [40,41,42]; for CRP, 18–19 h [43,44], and it is constant; the reported experimental animal half-life for *p*-cresol sulfate is up to 12 h [45]. Taking these data into account, we considered that a two-week wash-in period provides an appropriate time margin. A longer wash-out (four weeks) was chosen due to the fact that there may be some microbiota modifying the effect of ingested fiber, which may take a longer time (at least four weeks) to vanish [46]. The duration of the second phase with increased intake of fiber and sodium propionate was set at 8 weeks with the aim of reducing dysbiosis of the microbiota, and demonstrating a synergistic effect of MCO/HDF and fiber supplementation [47].

A clinical nutritionist will monitor the increased daily intake of dietary fiber. The target dietary fiber intake will be 40 g/day, with a minimum intake of 14 g/1000 kcal per day for patients who cannot reach the 40 g/day target, in accordance with the guidelines of European Society of Cardiology for Primary Prevention of Cardiovascular Disease and the American Institute of Medicine [48]. Fiber intake will be achieved by using natural whole grain sources in the habitual meal plan and a supplement of a dietary fiber mixture (psyllium 69% and inulin 30% blend) at a dose of 5 g BID for 8 weeks in the second phase of the study. The fiber supplement will also contain short-chain fatty acid sodium propionate (1 g/day), which will be added to a fiber supplement at the dose of 500 mg BID. It has been reported that short-chain fatty acids have a positive effect on systemic inflammation in patients on maintenance hemodialysis [49,50]. Short-chain fatty acids are a product of indigestible dietary carbohydrates, and we included sodium propionate to enhance the effect of dietary fiber supplementation, especially in patients who cannot achieve the desired goal of fiber intake. We do not expect any significant sodium-, potassium- or acid-based adverse effects from this dietary change [49].

Our study’s inclusion criteria will be the following: patients on chronic hemodialysis or hemodiafiltration for at least 12 weeks, age 18 years or more, a functioning arteriovenous fistula or graft as a permanent dialysis vascular access and being able to give informed consent to participate in the survey. Patients will not be included if planned for kidney transplantation, the transition to peritoneal dialysis or to another dialysis center within 12 weeks of the start of the study, patients with an episode of acute febrile illness 4 weeks prior to study inclusion, active chronic inflammation (e.g., an active autoimmune disease or an open wound), chronic ongoing infection or cancer, new cardiovascular or cerebrovascular event 4 weeks prior to study inclusion, clinically malnourished patients and/or BMI below 19 kg/m^2^ and/or loss of more than 5% of body mass in the last 3 months, immunosuppressive treatment, expected survival of less than 1 year, pregnancy or breast-feeding, indication for dietary supplements to increase calorie and/or protein intake, patients with specific indication for hemodiafiltration instead of hemodialysis as per attending physician, serum albumin concentration < 32 g/L at screening phase, inability to follow the study diet or test procedures, rapid reduction in residual renal function in the period before entry into the study and being intolerant of online hemodiafiltration (infusion intolerance).

During the dialysis period with the MCO membrane and during high-flux membrane hemodiafiltration, the following basic conditions of the dialysis procedure are planned: blood flow 300 mL/min; dialysate flow 500 mL/min; unchanged duration of dialysis procedures and the unaltered surface area of the dialysis membrane for the total study period; fluid removal by ultrafiltration according to the prescription of the treating physician and the clinical condition; for online hemodiafiltration, the target convection volume will be greater than or equal to 23 L per procedure.

All participants will be under the supervision of dialysis personnel, including the consultant nephrologist and experienced renal dialysis nurses. All adverse events will be prospectively monitored and reported by the study coordinator or principal investigator. 

### Sample Size

We considered that the expected standard deviation of IL-6 concentration in the HD population was 1.4 pg/mL [49] and estimated that the intraindividual correlation of IL-6 values was at least 0.5. The minimum value of the concentration difference that the study aims to detect is at least half the standard deviation, i.e., 0.7 pg/mL. A sample size of 31 patients is required for the alpha error value of 0.05 and the beta error of 0.2. In the expected dropout of 20% of patients in the 16-week period, a sample size of 39 patients is required, rounded up to 40 patients. When also considering the expected standard deviation of the *p*-cresol sulfate concentration of 0.13 mg/dL and the expected value of the difference between the two study arms of at least 0.11 mg/dL, at the value of alpha error of 0.05 and beta error of 0.2, a sample size of 44 patients is required. In consideration of the expected dropout, a sample size of 49 patients is required, rounded up to 50 patients, which is our target number for this study.

## 4. Conclusions

There are two main options for reducing the components of residual uremic syndrome in HD patients—improving purification techniques and reducing the formation and/or absorption of uremic toxins. In recent years, innovation in dialysis membrane technology has been introduced into clinical practice with the development of MCO membranes. In our systematic review, we could not find any studies investigating the effect of dialysis with MCO membrane on PBUT values. The effects of MCO membrane dialysis on inflammation markers were inconsistent and inconclusive. On the other hand, we found compelling evidence from recent studies on the use of dietary fiber supplements to reduce inflammation levels and PBUT concentrations. Therefore, a study that investigates both therapeutic approaches in combination would be justified to answer this question. However, it is premature to conclude the superiority of the combined intervention at this time. Should this combination prove to be effective and safe, it could represent an important step towards improving the important components of residual uremic syndrome in HD patients and possibly contribute to improving their prognosis.

## 5. Materials and Methods 

### 5.1. Literature Search Strategy

For this systematic review, PRISMA guidelines were followed [51]. Our main objective was to find randomized controlled interventional studies that investigated the effects of MCO membranes and/or dietary fiber on PBUT concentration and/or inflammatory markers. The electronic databases investigated were PubMed, Cochrane, Embase and Web of Science from 15 January 2021 to 12 February 2021. The following keywords were used to search for the articles reporting on the performance of medium cut-off dialyzers compared to hemodiafiltration or high-flux HD: “medium cut-off membrane” OR “medium cut-off dialyser” OR “expanded hemodialysis” OR “expanded haemodialysis” OR “high-flux hemodialysis” OR “high-flux haemodialysis” AND “hemodiafiltration” OR “haemodiafiltration” AND “protein-bound uremic toxins” OR “protein-bound uremic solutes” OR “indoxyl sulfate” OR “3-indoxyl sulfate” OR “3-indoxyl sulfuric acid” OR “indol-3-yl sulfate” OR “*p*-cresol” OR “p-cresyl” OR “4-cresol” OR “4-cresyl” OR “p-cresol sulfate” OR “p-cresyl sulfate” OR “4-cresol sulfate” OR “4-cresyl sulfate” OR “inflammation” OR “inflammation markers” OR “inflammation parameters” OR “inflammatory markers” OR “inflammatory parameters” OR “CRP” OR “hs-CRP” OR “IL-6” OR “serum amyloid A” OR “SAA”.

The keywords in the search for articles on the influence of dietary fiber supplementation on serum concentrations of PCS, IS and inflammation markers were “hemodialysis” OR “haemodialysis” OR “end-stage kidney disease” OR “end-stage renal disease” AND “fiber” OR “fibre” OR “dietary fiber” OR “dietary fibre” OR “nondigestible fiber” OR “nondigestible fibre” OR “prebiotic” OR “starch” OR “resistant starch” AND “indoxyl sulfate” OR “3-Indoxylsulfate” OR “3-Indoxylsulfuric acid” OR “Indol-3-yl sulfate” OR “*p*-cresol” OR “*p*-cresyl” OR “4-cresol” OR “4-cresyl” OR “*p*-cresol sulfate” OR “*p*-cresyl sulfate” OR “4-cresol sulfate” OR “4-cresyl sulfate” OR “protein-bound uremic toxins” OR “protein-bound uremic solutes” OR “IL-6” OR “CRP” OR “hs-CRP” OR “serum amyloid A” OR “SAA” OR “inflammation” OR “inflammation markers” OR “inflammation parameters” OR “inflammatory markers” OR “inflammatory parameters“.

The literature search was performed independently by two authors (T.H. and T.V.). First, the authors carried out a screening of the titles. Second, the abstracts were selected for further analysis on the basis of previously agreed inclusion and exclusion criteria. Finally, the full text of the included papers was read. In the case of disagreement between the authors, the issue was clarified by a corresponding author (J.P.). If the full text of the paper was not available or some information was missing, we contacted the corresponding and/or the first author by e-mail or via the Research Gate platform. The progress of the screening is shown in Figure 4.

### 5.2. Inclusion and Exclusion Criteria

The present review included only original articles in English published in peer-reviewed journals. Inclusion criteria were the following: randomized controlled trials in HD patients, studies that investigated the effect of the MCO dialyzer on PBUT and inflammatory marker concentrations or studies that investigated the influence of dietary fiber supplementation on the same endpoints. Due to the small number of eligible studies, we did not include criteria for study duration. Exclusion criteria were non-HD patients, in vitro and animal studies, systematic reviews and meta-analyses, conference abstracts, theses and case reports.

### 5.3. Quality Assessment

Two authors independently (T.V. and T.H.) assessed the methodological quality and risk of bias of included studies. To assess the methodological quality of the included studies, we used the Physiotherapy Evidence Database (PEDro) scale [52]. The PEDro scale consists of 11 items that assess the methodological quality of a study and can be used in various fields of medicine. Each scale item contributes 1 point to the total PEDro score (0–10 points). The total score is interpreted as follows: ≤3 points (poor methodological quality), 4–5 points (moderate quality) and 6–10 points (high quality). Agreement between the two reviewers was assessed using k-statistics. In the case of disagreement, a corresponding author (J.P.) made the final decision.

## Figures and Tables

**Figure 1 toxins-13-00244-f001:**
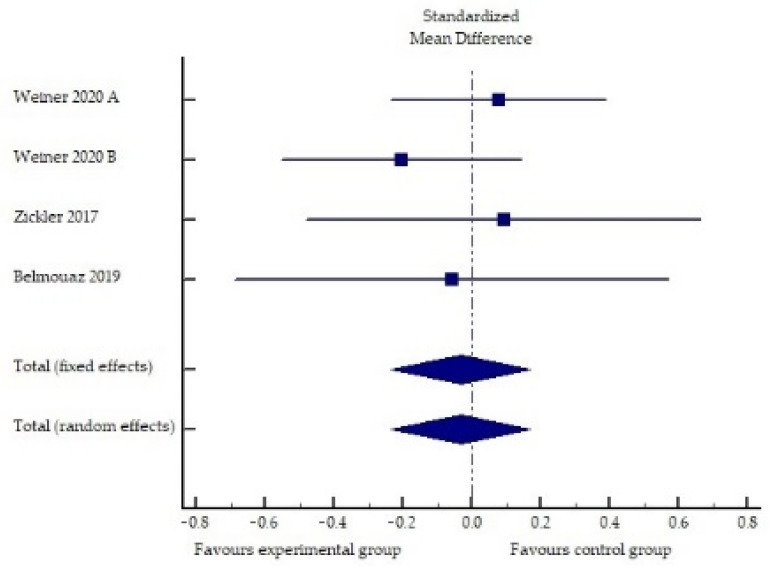
Forest plot of the standardized mean differences of the MCO membrane effect on interleukin 6.

**Figure 2 toxins-13-00244-f002:**
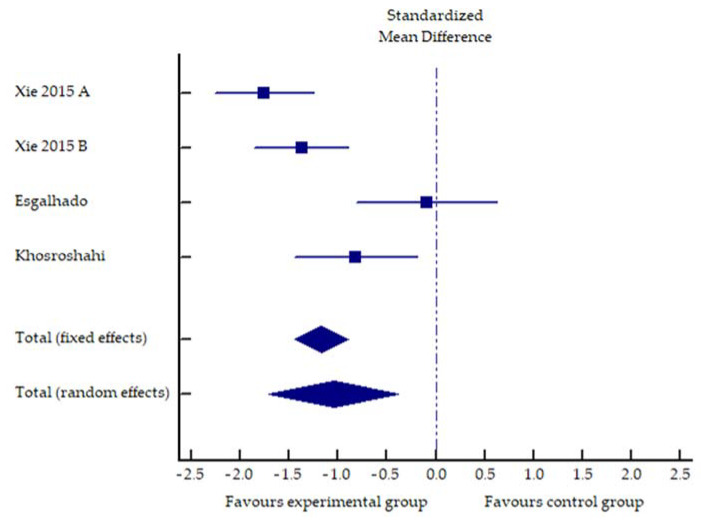
Forest plot of the standardized mean differences of the fiber intake effect on interleukin 6. Experimental group—fiber supplementation. Control group—various forms of fiberless carbohydrates.

**Figure 3 toxins-13-00244-f003:**
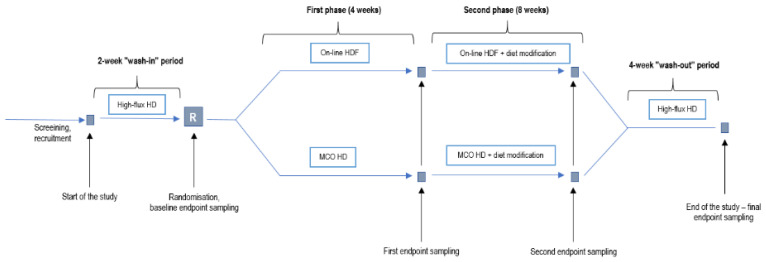
Study flow diagram. Abbreviations: MCO, medium cut-off membrane (Theranova); HD, hemodialysis; HDF, online hemodiafiltration.

**Figure 4 toxins-13-00244-f004:**
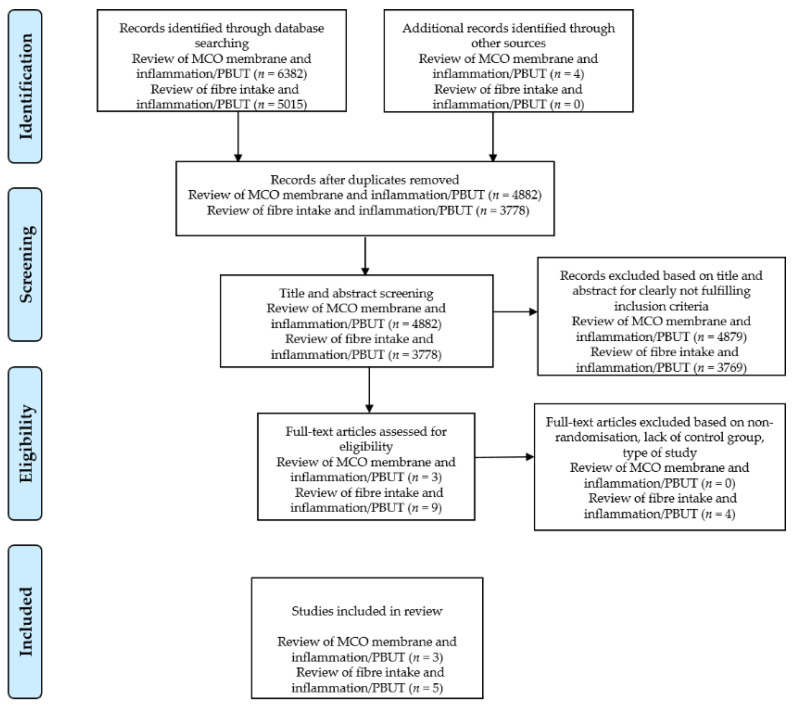
PRISMA flow diagram. Abbreviations: MCO, medium cut-off membrane (Theranova); PBUT, protein-bound uremic toxins.

**Table 1 toxins-13-00244-t001:** Systematic review summary of the impact of hemodialysis (HD) with medium cut-off membrane (MCO) on inflammatory markers.

Study	Sample Size	Exp Intervention	Con Intervention	Duration	Endpoint	Results
Weiner, 2020 [36]	Exp (*n* = 86) Con (*n* = 86)	MCO membrane	High-flux HD	24 weeks	IL-6	Exp: 15% ↓
						Con: 50.6% ↑
						p Exp vs. Con n.s.
					TNF-α	Exp: 48.9% ↓
						Con: 34.7% ↓
						p Exp vs. Con n.s.
Zickler, 2017 ^†^ [37]	Exp (*n* = 23) Con (*n* = 25)	MCO membrane	High-flux HD	4 weeks of each dialysis modality + 8 weeks of extension phase	IL-6	Exp: 33.3% ↓ **
						Con: 43.9% ↓ *
						p Exp vs. Con n.s.
					TNF-α	Exp: 14.5% ↓ **
						Con: 5.1% ↓
						p Exp vs. Con n.s.
					CRP	Exp: 39.2% ↓
						Con: 28.4% ↓
						p Exp vs. Con n.s.
Belmouaz, 2020 [38]	Exp (*n* = 20) Con (*n* = 20)	MCO membrane	High-flux HD	3 months of each dialysis modality	IL-6	Exp: 6.3% ↑
						Con: 12.8% ↑
						p Exp vs. Con n.s.
					TNF-α	Exp: 19.4% ↓
						Con: 16.4% ↓
						p Exp vs. Con n.s.
					CRP	Exp: missing data ^1^
						Con: missing data ^1^
						p Exp vs. Con n.s.

Abbreviations: Exp, experimental group; Con, control group; * *p* < 0.05 within group; ** *p* < 0.01 within group; RR, reduction ratio; MCO, medium-cut off; HD, hemodialysis; IL-6, interleukin 6; TNF-α, tumor necrosis factor α; CRP, C-reactive protein; ^1^ only postdialysis values. ^†^ IL-6 mRNA significantly reduced in Exp group (23.1%) with significant differences between groups (*p* < 0.001); TNF-α mRNA significantly reduced in both groups (Exp: 18.5%; Con: 14.3%) with significant differences between groups (*p* < 0.001).

**Table 2 toxins-13-00244-t002:** Systematic review summary of the impact of dietary fiber on protein-bound uremic toxins and inflammatory markers.

Study	Sample Size	Exp Intervention	Con Intervention	Duration	Endpoint	Results
Khosroshahi, 2019 [18]	Exp (*n* = 23) Con (*n* = 21)	20–25 g of HAM-RS2 20 g week 1–4 25 g week 5–8	20–25 g of waxy corn starch 20 g week 1–4 25 g week 5–8	8 weeks		Exp: 31.5% ↓ *
					PC	Con: 0.7% ↑
						p Exp vs. Con 0.992
						Exp: 2.96% ↑
					IS	Con: 11.7% ↑
						p Exp vs. Con 0.606
						Exp: 110.6% ↑
						Con: 25.05% ↑
					hs-CRP	p Exp vs. Con 0.866
Esgalhado, 2018 [19]	Exp (*n* = 15) Con (*n* = 16)	16 g HAM-RS2	20 g manioc flour	4 weeks		Exp: 18.15% ↓ **
					IS	Con: 9.28% ↑
						p Exp vs. Con 0.008
						Exp: 6.83% ↑
						Con: 1.53% ↓
					PCS	p Exp vs. Con 0.77
						Exp: 11.86% ↓ **
						Con: 3.28% ↓
					IL-6	p Exp vs. Con 0.06
						Exp: 20.0 ↓
						Con: 12.77 ↓
					hs-CRP	p Exp vs. Con 0.16
Sirich, 2014 [20]	Exp (*n* = 20) Con (*n* = 20)	15 g HAM-RS2 week 1, 30 g HAM-RS2 week 2–6	15 g waxy corn starch week 1, 30 g waxy corn starch week 2–6	6 weeks		Exp: 30.56% ↓ *
					free IS	Con: 0%
						p Exp vs. Con 0.02
						Exp: 17.14% ↓
						Con: 3.12% ↓
					total IS	p Exp vs. Con 0.04
						Exp: 22.22% ↓
						Con: 4.55% ↑
					free PCS	p Exp vs. Con 0.05
						Exp: 12.12 % ↓
						Con: 3.12% ↓
					total PCS	p Exp vs. Con 0.63
						Exp: 10% ↑
						Con: 33.33 % ↑
					CRP	p Exp vs. Con 0.11
Xie, 2015 [22]	ExpA (*n* = 41) ExpB (*n* = 39) Con (*n* = 44)	ExpA: 10 g water soluble fiber ExpB: 20 g water soluble fiber	placebo starch	6 weeks		ExpA: 32.63% ↓ *
						ExpB: 27.22% ↓ *
					IL-6	Con: 6.19% ↑
						p ExpA vs. Con < 0.05
						p ExpB vs. Con < 0.05
						ExpA: 76.58% ↓ *
						ExpB: 62.25% ↓ *
						Con: 10.05% ↑
					IL-8	p ExpA vs. Con < 0.05
						p ExpB vs. Con < 0.05
						ExpA: 24.06% ↓ *
						ExpB: 15.2% ↓ *
						Con: 0.77% ↑
					TNF-α	p ExpA vs. Con < 0.05
						p ExpB vs. Con < 0.05
						ExpA: 55.14% ↓ *
						ExpB: 52.04% ↓ *
						Con: 1.06% ↑
						p ExpA vs. Con < 0.05
					hs-CRP	p ExpB vs. Con < 0.05
Khosroshahi, 2018 [21]	Exp (*n* = 22) Con (*n* = 22)	20–25 g of HAM-RS2 20 g week 1–4 25 g week 5–8	20–25 g regular wheat-flour 20 g week 1–4 25 g week 5–8	8 weeks		Exp: 25.0% ↓ *
					IL-6	Con: 21.9% ↑
						p Exp vs. Con < 0.01
						Exp: 6.49% ↓
						Con: 20.0% ↑
					IL-1β	p Exp vs. Con. n.s.
						Exp: 18.23% ↓ *
						Con: 43.14% ↑ ***
						p Exp vs. Con 0.01
					TNF-α	
						Exp: 21.6% ↑
						Con: 1.27% ↑
					hs-CRP	p Exp vs. Con n.s.

Abbreviations: Exp, experimental group; Con, control group; IL-6/8/1β, interleukin 6/8/1β; TNF-α, tumor necrosis factor α; (hs) CRP (high-sensitivity), C-reactive protein; HAM-RS2, high amylose resistant starch (HAM-RS2); PC, *p*-cresol; PCS, *p*-cresol sulphate; IS, indoxyl sulfate; * *p* < 0.05 within group; ** *p* < 0.01 within group; *** *p* < 0.001 within group; n.s., non-significant.

**Table 3 toxins-13-00244-t003:** Physiotherapy Evidence Database (PEDro) scale of included studies.

Study	Criterion 1	Criterion 2	Criterion 3	Criterion 4	Criterion 5	Criterion 6	Criterion 7	Criterion 8	Criterion 9	Criterion 10	Criterion 11	Score
Weiner 2020	1	1	0	1	0	0	1	0	1	1	1	6
Zickler 2017	1	1	1	1	0	0	1	1	1	1	1	8
Belmouaz 2020	1	1	1	1	0	0	0	0	0	1	1	5
Khosroshahi 2019	1	1	0	1	1	1	1	1	1	1	1	9
Esgalhado 2018	1	1	0	1	1	1	1	0	0	1	1	7
Sirich 2014	1	1	0	1	1	0	0	0	0	1	1	5
Xie 2015	1	1	0	1	1	0	0	1	1	1	1	7
Khosroshahi 2018	1	1	0	1	1	1	1	1	0	1	1	8

Criterion 1: eligibility criteria specified; Criterion 2: random allocation; Criterion 3: concealed allocation; Criterion 4: baseline group similarity; Criterion 5: participant blinding; Criterion 6: personnel blinding; Criterion 7: assessor blinding; Criterion 8: more than 85% participants measured for the key endpoint; Criterion 9: intention to treat analysis; Criterion 10: between-group comparison conducted; Criterion 11: point measures /measurement variability. (All criteria except Criterion 1 contribute to the final score.)

## Data Availability

Not applicable.

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
