# Peer review of "Medium Cut-Off Dialysis Membrane and Dietary Fiber Effects on Inflammation and Protein-Bound Uremic Toxins: A Systematic Review and Protocol for an Interventional Study"

_toxins, 2021, doi:10.3390/toxins13040244_

Round 1
Reviewer 1 Report
The authors review the effects of medium cut-off membrane dialysis (MCO HD) and dietary fiber (DF) on inflammation and protein-bound uremic toxins and proposed a protocol for an interventional study. The review of the MCO HD and DF is well written. However, I have several comments, which could improve the impact of this manuscript.
The mechanism that involved in the results of the study is poorly discussed. The study design that could influence to the results, such as patient characteristics and dialysis condition, should be further discussed.
The proposed interventional study seems to be complicated design to answer the hypothesis. Why did you compare exchange from High-flux HD to MCO HD or On-line HDF? It would be simple to compare exchange to MCO HD or keep on High-flux HD. Comparison between MCO HD and On-line HDF seem to be next study. In addition, is 4 weeks observation enough period from the data in the previous study?
Author Response
"Please see the attachment."

Reviewer 2 Report
Dear authors,
I read your manuscript with great interest. I appreciate the importance of the presented topic as well as its relevance to the clinical procedures in AKI units.
On the other hand, I feel that some crucial mistakes have been made during the searching/planning stage of the presented study. I am aware that my comments might be not as kind as expected, but I deeply hope that they will lead to substantial improvement in the paper, that finally might be suitable to publish in Toxins.
Major:
- Lines 98-99: you present a very broad point of interest. You combine key-players of CKD/AKI/EDRD: UTs, fiber Dialysis, and inflammatory markers. It limits per se your power of the statistical approach.
- Line 100. I do not understand why it took so long to search the databases. Selections of potential papers for systematic review should be fast and precise to eliminate additional biases.
- I do not fully agree with the chosen keywords. Middle molecule, inflammation, or starch give you lots of false-positive searching results. You put so many different names for p/cresyl but only one for IS - keep in mind that IS (even for me!) has different names - 3-Indoxylsulfate; 3-Indoxylsulfuric acid; Indol-3-yl sulfate. It introduced high bias to your search strategy. CRP is not the same as hs-CRP. Prebiotic should be also expanded - symbiotic (pre+pro). There are many other TRP metabolites that might be interesting from the point of view of the paper. I feel (maybe I am wrong) that your strategy of searching (that is critical of systematic review) was highly biased.
- To continue point 3 - the result of a biased searching strategy is the number of papers that fulfill your criteria. Given numbers are very low from my point of view: n=74; n=32 etc. I also do not know how is it possible to find among 74 papers 16 duplicates. It shows that the overall strategy was wrong.
- Moving forward, I do not think that based on a few papers with a very low number of enrolled subjects authors were able to withdraw any scientifically sounded conclusions. It was literally gambling - one more paper might change all the data. The timepoints, analyzed parameters, sample size together makes the paper very flawed.
What I would do if I were you: The topic is important, the data are conflicting in many papers, people working in ICU/AKI units need more evidence-based medicine. So, I would rather focus on membranes/UT or fiber/inflammatory markers. Do not mix them. Please re-search the databases. Honestly, I am aware of how much work it is but in the current form the paper can not be accepted. Moreover, I would rather focus on the review, not on the protocol that was not necessary for a paper like this.
The systematic review is always serious work that starts at the search strategy. If it falls, everything falls down.
Please accept my apologies for being rude but I hope to review the revised version of the paper.
Best.
Author Response
"Please see the attachment."

Round 2
Reviewer 2 Report
Dear Authors,
Thank you for your careful review and the changes that have been incorporated into the manuscript.
For sure the draft has been substantially improved, but I am not sure if the paper can be published in the present form. My main concern focuses on the relatively low number of papers included in the study. I am not sure if the conclusions are supported by the endpoints of the analysis. It looks like there are 65% that your conclusions are genuine and 35% that you made a mistake when withdrawing the final conclusions for the paper.
I think how you can minimize the risk of rejection and stay on the safe side: Please carefully discuss the limitations of this study - a new paragraph should be a wise attempt to ensure the readers that you are aware of the existence of the potential bias.
Please take this advice seriously and create this new paragraph - it will make your manuscript definitely more suitable for publication.
Best,
Author Response
"Please see the attachment."

Round 3
Reviewer 2 Report
Dear Authors,
I am really happy with all the changes you made. I feel that the only one thing that might be improved is right now the section describing the influence of PBUT that starts at line 90. Since there is a plethora of evidence (recently published) on how IS/p-cresol etc. impact inflammation and oxidative stress in clinical conditions the new findings should be also incorporated, these papers might be handy:
doi: 10.3390/toxins10090367
DOI: 10.1111/hdi.12483
https://doi.org/10.1161/JAHA.116.005022
https://doi.org/10.1186/s12882-017-0457-1
Moreover, please double-check all the spellings and references.
Best,
Author Response
Dear editors and reviewers,
many thanks for your additional advice on the manuscript. We have followed your advice and incorporated suggested changes describing the influence of PBUTs. Expanded text can now be found in the 4th paragraph of the introduction section as follows:
"In vitro studies [24–27] and studies in CKD patients [28] suggested an association of some PBUTs with inflammatory and oxidative stress markers. Rossi et al. [28] demonstrated a significant correlation of free and total IS with interleukin-6 (IL-6), interferon-gamma (IFN-g), and tumor necrosis factor-alpha (TNF-a) as well as a correlation of free and total PCS with IL-6. In vitro studies showed the ability of PCS and IS to stimulate gene expression of IL-6 [24] and IS to stimulate expression of TNF-a [25]. PCS and IS induced reactive oxygen species, which activate the nuclear factor-kappaB (NF-kb) signaling pathway, leading to oxidative stress as well as the production of proinflammatory cytokines [26,27]. Besides inflammation additional effects of PCS and IS are related to free radical production, and fibrosis [29]. PCS and IS are both associated with renal tubulointerstitial fibrosis and cardiovascular disease progression [29,30]. IS has been associated with smooth muscle cell proliferation and vascular calcification, and thus with the progression of atherosclerosis with increased aortic stiffness and the progression of peripheral arterial disease. Through the mechanism of cardiac hypertrophy and fibrosis, increased IS levels have been associated with congestive heart failure and arrhythmia in CKD patients [31–33]. IS has been associated with thrombosis in vascular accesses in HD patients [31] and has also been associated with disorders in the hemostatic system [34] and renal bone disease in CKD patients [32,35]. Elevated PCS has been associated with insulin resistance and the development of metabolic syndrome and its complications in CKD patients [30]. All in all, these data show a plethora of evidence for the influence of PBUT on various factors leading to increased morbidity in dialysis patients."
We hope that you will find the revisions satisfactory. Very best regards,